# RAC1B Suppresses TGF-β-Dependent Chemokinesis and Growth Inhibition through an Autoregulatory Feed-Forward Loop Involving PAR2 and ALK5

**DOI:** 10.3390/cancers11081211

**Published:** 2019-08-20

**Authors:** Hannah Otterbein, Koichiro Mihara, Morley D. Hollenberg, Hendrik Lehnert, David Witte, Hendrik Ungefroren

**Affiliations:** 1First Department of Medicine, University Hospital Schleswig-Holstein, Campus Lübeck, D-23538 Lübeck, Germany; 2Departments of Physiology and Pharmacology and Medicine, Inflammation Research Network, Snyder Institute for Chronic Diseases, University of Calgary, Cumming School of Medicine, Calgary, AB T2N 4N1, Canada; 3Clinic for General Surgery, Visceral, Thoracic, Transplantation and Pediatric Surgery, University Hospital Schleswig-Holstein, Campus Kiel, D-24105 Kiel, Germany

**Keywords:** RAC1B, ALK5, PAR2, cell migration, pancreatic carcinoma, RNA interference, TGF-β

## Abstract

The small GTPase RAC1B functions as a powerful inhibitor of transforming growth factor (TGF)-β1-induced epithelial-mesenchymal transition, cell motility, and growth arrest in pancreatic epithelial cells. Previous work has shown that RAC1B downregulates the TGF-β type I receptor ALK5, but the molecular details of this process have remained unclear. Here, we hypothesized that RAC1B-mediated suppression of activin receptor-like kinase 5 (ALK5) involves proteinase-activated receptor 2 (PAR2), a G protein-coupled receptor encoded by *F2RL1* that is crucial for sustaining ALK5 expression. We found in pancreatic carcinoma Panc1 cells that PAR2 is upregulated by TGF-β1 in an ALK5-dependent manner and that siRNA-mediated knockdown of RAC1B increased both basal and TGF-β1-induced expression of PAR2. Further, the simultaneous knockdown of PAR2 and RAC1B rescued Panc1 cells from a RAC1B knockdown-induced increase in ALK5 abundance and the ALK5-mediated increase in TGF-β1-induced migratory activity. Conversely, Panc1 cells with stable ectopic expression of RAC1B displayed reduced ALK5 expression, an impaired upregulation of PAR2, and a reduced migratory responsiveness to TGF-β1 stimulation. However, these effects could be reversed by ectopic overexpression of PAR2. Moreover, the knockdown of PAR2 alone in Panc1 cells and HaCaT keratinocytes phenocopied RAC1B’s ability to suppress ALK5 abundance and TGF-β1-induced chemokinesis and growth inhibition. Lastly, we found that the RAC1B knockdown-induced increase in TGF-β1-induced PAR2 mRNA expression was sensitive to pharmacological inhibition of MEK-ERK signaling. Our data show that in pancreatic and skin epithelial cells, downregulation of ALK5 activity by RAC1B is secondary to suppression of *F2RL1*/PAR2 expression. Since *F2RL1* itself is a TGF-β target gene and its upregulation by TGF-β1 is mediated by ALK5 and MEK-ERK signaling, we suggest the existence of a feed-forward signaling loop involving ALK5 and PAR2 that is efficiently suppressed by RAC1B to restrict TGF-β-driven cell motility and growth inhibition.

## 1. Introduction

RAC1B is an alternative splice product derived from the human *RAC1* gene. This isoform differs from RAC1 by an in-frame insertion of 57 nucleotides, comprising an additional exon (exon 3b) [1]. RAC1B can promote cell cycle progression and apoptosis resistance, however, its role in other processes involved in driving tumor progression, like epithelial–mesenchymal transition (EMT), cell motility, and metastasis is less well known. The inclusion of exon 3b not only impairs hydrolysis of GTP and accelerates exchange of GDP to GTP, but also confers specific biochemical and signaling properties on RAC1B that eventually result in specific cellular responses different from those of Rac1 ([1] and references therein). For instance, we showed earlier in pancreatic ductal adenocarcinoma (PDAC) that RAC1B suppresses EMT, random cell migration, and growth inhibition induced by TGF-β1, while RAC1 promotes both processes [2,3,4,5]. This was in contrast to the promoting effect of Rac1b on EMT induced by MMP3 in SCp2 murine mammary epithelial cells [6,7]. Previous data from our group also indicated that RAC1B suppresses expression and TGF-β1-mediated upregulation of activin receptor-like kinase 5 (ALK5), the prototype type I receptor for TGF-β [8]. This suppression was associated with decreased activation of Smad [2], MKK3/6-p38, and MEK-ERK signaling [3]. However, the molecular mechanism(s) underlying RAC1B-mediated inhibition of ALK5 expression and function remains obscure, i.e., whether regulation is executed at the level of the *TGFBR1* promoter to prevent de novo transcription, or whether the RAC1B effect on ALK5 is indirect via other mechanisms.

Previously published data have shown that *F2RL1*, encoding the G protein-coupled receptor proteinase-activated receptor 2 (PAR2), is upregulated by TGF-β1 and that this upregulation is strongly enhanced upon reducing the abundance of RAC1B via small interfering RNA (siRNA)-mediated knockdown [3]. Since PAR2 protein is strictly required for sustaining expression of ALK5 [9], although its tethered-ligand-mediated activation is not required for promoting TGF-β1/ALK5-stimulated cell migration [10], we considered the possibility that RAC1B can inhibit PAR2 upregulation, which subsequently causes a decline in PAR2 expression and thereby attenuates TGF-β1/ALK5-dependent cellular responses. To test this hypothesis, we employed as the primary model system Panc1 cells with transient RAC1B knockdown, CRISPR/Cas-mediated deletion of exon 3b of *RAC1* (RAC1B-KO) [8], or stable ectopic overexpression of RAC1B [2] to validate the following scenario: (i) RAC1B suppresses basal and TGF-β-induced PAR2 expression through inhibition of specific signaling pathways, (ii) reduced synthesis of PAR2 results in lower abundance ALK5, (iii) lower levels of ALK5 cause impaired signaling upon TGF-β1 challenge, and (iv) an impaired signaling response to TGF-β will ultimately result in less production of PAR2. This series of events may generate a circulus vitiosus to efficiently suppress TGF-β/ALK5-dependent cellular responses, i.e., cell migration and growth inhibition. 

## 2. Results

### 2.1. TGF-β-Dependent PAR2 Expression Is Negatively Regulated by RAC1B in an ALK5-Dependent Manner

We have previously shown that PAR2 is required for sustaining ALK5 expression in PDAC-derived Panc1 and Colo357 cells, as well as in skin-derived HaCaT cells [9], all cell lines in which TGF-β-induced migration and cell cycle arrest had been well characterized [2,3,8,9,10,11,12]. *F2RL1* itself is upregulated in Panc1 cells following treatment with the recombinant mature form of TGF-β1 [3]. Induction of PAR2 mRNA by TGF-β1 was also noted in BxPC3 and Colo357 cells, but not in TGF-β type II receptor-deficient MiaPaCa2 cells (Appendix A). Given the potent inhibitory effect of RAC1B on ALK5 expression and TGF-β/ALK5-mediated signaling [2,3,8] we sought to know if RAC1B also regulates TGF-β-induced expression of *F2RL1*. Initially, we performed an siRNA-mediated knockdown of RAC1B in Panc1 cells and determined the effects on TGF-β1-induction of PAR2 mRNA by qPCR analysis. While in control siRNA-transfected Panc1 cells, PAR2 was induced 5.4-fold after 24 h of ligand treatment, and induction was strongly enhanced to 9-fold over control in RAC1B siRNA transfected cells (Figure 1A). Interestingly, the strong increase in TGF-β1 induction of PAR2 mRNA following RAC1B knockdown could be reversed by concomitant knockdown of ALK5 (Figure 1B), or ectopic expression of a kinase-dead ALK5 mutant (K232R, data not shown).

Next, we evaluated the effects of TGF-β1 on the abundance of *F2RL1* mRNA in individual clones of Panc1 cells with stable ectopic overexpression of HA-tagged RAC1B (HA-RAC1B). The generation and migratory behavior of these clones has been characterized previously [2]. As shown in Appendix A and Figure 1C, in all individual Panc1-HA-RAC1B clones, there was a downregulation in the abundance of ALK5 mRNA and protein, respectively, and in the ability of TGF-β1 to induce *F2RL1* mRNA (Figure 1D), as compared to empty vector control cells. Finally, we asked whether reintroduction of HA-RAC1B into cells depleted of endogenous RAC1B by CRISPR/Cas technology (Panc1-RAC1B-KO) can rescue these cells from experiencing the increase in ALK5 expression described earlier [8]. To this end, transfection of Panc1-RAC1B-KO cells with HA-RAC1B but not empty vector led to a strong decrease in the expression of both ALK5 protein and PAR2 mRNA as well as in TGF-β1-induced PAR2 mRNA induction (Appendix A).

The protein abundance of endogenous PAR2 could not be visualized and, hence, was assessed by immunoblotting due to the lack of sensitive PAR2 antibodies ([13,14] and our own unpublished results). Nevertheless, the data show that *F2RL1* is a TGF-β1 target gene in Panc1 cells and that its expression is positively controlled by ALK5 and negatively by RAC1B.

### 2.2. The RAC1B Knockdown-Induced Upregulation of ALK5 Is PAR2-Dependent but PAR2 Upregulation in the RAC1B Knockdown Cells Is ALK5-Independent 

Our studies have shown that both *TGFBR1* (encoding ALK5) [8] and *F2RL1* ([3] and Figure 1) are TGF-β target genes in Panc1 cells. With regard to *TGFBR1*, RAC1B suppresses both its basal expression as well as the response to its bona fide ligand, TGF-β1 [8], however this inhibitory effect of RAC1B does not appear to involve decreased transcriptional activity from the *TGFBR1* promoter (H.U., unpublished observation). We therefore hypothesized that rather than inhibiting the expression of the ALK5 gene directly, RAC1B may target ALK5 through downregulation of PAR2 and, as a consequence of reducing ALK5, attenuate TGF-β signaling and TGF-β-dependent cellular responses. More specifically, we postulated that cells depleted of PAR2 will fail to increase ALK5 in response to a RAC1B knockdown. To this end, cotransfection of Panc1 cells with RAC1B siRNA plus PAR2 siRNA relieved the stimulatory effect of the RAC1B knockdown on basal levels of ALK5 mRNA (Figure 2A) and protein (Figure 2B), as well as on the small induction of *TGFBR1* by recombinant TGF-β1 seen in RAC1B-knockdown cells (Figure 2C) but not the corresponding RAC1B-expressing control cells [8]. Conversely, knocking down ALK5 by siRNA did not affect the RAC1B knockdown-induced increase in basal PAR2 expression (Figure 2D), indicating that PAR2 controls ALK5 expression but that ALK5 was not required for an increase in PAR2 mRNA, as is the case for TGF-β1-induced PAR2 upregulation (Figure 1B). Finally, to confirm the results from the RAC1B knockdown approach, we knocked down PAR2 in Panc1-RAC1B-KO cells. As predicted by our model, ALK5 abundance was dramatically reduced in PAR2 siRNA-transfected RAC1B-KO cells (Figure 2E). Together, these data reveal the existence of a positive feedback loop with mutual regulation of ALK5 and PAR2 in response to TGF-β1 that is normally suppressed by RAC1B. Upon RAC1B depletion and TGF-β1 stimulation, the mutual stimulatory effects between ALK5 and PAR2 are freed from inhibition, eventually resulting in upregulation of ALK5 and TGF-β/ALK5 signaling.

### 2.3. The RAC1B Knockdown-Mediated Increase in TGF-β-induced Migration Is Reversed by PAR2 Knockdown while the RAC1B Overexpression-Mediated Decrease in TGF-β-Induced Migration Is Reversed by Ectopic PAR2 Expression 

Given the crucial role of PAR2 in enabling the upregulation of ALK5 in RAC1B-knockdown cells (see Figure 2), we postulated that knockdown of PAR2 will block the increase of ALK5 in RAC1B-depleted cells, thereby preventing an increase in TGF-β1-dependent random cell migration in the RAC1B-knockdown cells. To this end, PAR2 siRNA transfection completely and partially, respectively, prevented the RAC1B depletion-mediated upregulation of TGF-β1-induced chemokinesis in Panc1-RAC1B-knockdown (Figure 3A) and -KO (Appendix A) cells. Moreover, the knockdown of PAR2 alone (in the absence of reduced endogenous RAC1B expression) was able to mimic the inhibitory effect of RAC1B on TGF-β1-induced cell migration in HaCaT cells (Appendix A). This result strongly suggests that RAC1B downregulates PAR2 to reduce the expression of ALK5 and thereby impairs TGF-β1/ALK5-mediated cell migration.

Above, we have shown that ectopic expression of RAC1B inhibited ALK5 expression as well as TGF-β1/ALK5-mediated upregulation of PAR2 (see Figure 1C,D). Since ectopic RAC1B also inhibited the cells’ chemokinetic response to TGF-β1 [2,3], we hypothesized that the migration-inhibitory effect of RAC1B is due to downregulation of *F2RL1* and should be reversed following ectopic overexpression of PAR2 protein. In keeping with this hypothesis, ectopic PAR2 expression partially rescued Panc1-HA-RAC1B cells from the antimigratory effect of RAC1B overexpression (Figure 3B). In sum, both experimental approaches, PAR2 knockdown and ectopic overexpression, point to PAR2 as being a centrally involved participant in RAC1B-induced suppression of TGF-β1-dependent chemokinesis.

### 2.4. PAR2 Knockdown Protects Cells from the RAC1B Knockdown-Mediated Increase in TGF-β-Induced Growth Inhibition

We have previously shown in various cell types that RAC1B antagonizes the growth-inhibitory effect of TGF-β1 [2,3,8] due to downregulation of ALK5 [8]. If both events were mediated by intermittent downregulation of PAR2 expression, then knockdown of PAR2 should prevent endogenous RAC1B from antagonizing TGF-β1-dependent growth arrest. To validate this assumption, we transfected Panc1 and HaCaT cells with either PAR2 siRNA or irrelevant control siRNA and determined cell counts after 24 h of TGF-β1 stimulation. Interestingly, *F2RL1* silencing was able to relieve the growth-inhibitory effect of TGF-β1 stimulation (Figure 4A), which was associated with a downregulation of ALK5 as determined by immunoblot analysis (Figure 4B). Similar results after PAR2 knockdown were obtained in HaCaT cells (Appendix A). In both cell types, cell viabilities were not different between control and PAR2 knockdown cells. Together, these data show that downregulation of PAR2 alone is sufficient for the observed decrease in ALK5 abundance which, in turn, allows cells to escape from growth/cell cycle inhibition by TGF-β1.

### 2.5. TGF-β1-Dependent Upregulation of F2RL1 Following RAC1B Knockdown Is Sensitive to Inhibition of MEK-ERK Signaling

We have shown earlier that in PDAC-derived cells RAC1B decreases phosphorylation (reflecting activation) of Smad2/3 [2] and non-Smad pathways (p38 and ERK mitogen-activated protein kinases [3]) in response to TGF-β1. To study if p38 or ERK signaling is involved in the RAC1B-mediated suppression of the *F2RL1* response to TGF-β1, we treated Panc1 cells with TGF-β1 in the presence or absence of pharmacological inhibitors of p38 (SB203580), MEK (U0126), or the ALK5 kinase (SB431542, used here as positive control). The qPCR-based analysis revealed that upregulation of PAR2 mRNA in RAC1B siRNA transfected cells was potently inhibited by U0126 and SB431542, but not SB203580 (Figure 5). From these data, we conclude that TGF-β1-induced upregulation of PAR2 mRNA is derepressed by knockdown of RAC1B and that RAC1B quenches the response of *F2RL1* to TGF-β1 by suppressing MEK-ERK signaling.

## 3. Discussion

In previous reports, we have demonstrated negative regulation of TGF-β1-dependent cell migration/chemokinesis by RAC1B in both benign and malignant pancreatic ductal epithelial cells. In the present study, we sought to unravel the mechanistic basis of this initially unexpected role of RAC1B, using pancreatic carcinoma Panc1 cells and HaCaT keratinocytes. Based on the observation that both transient knockdown of RAC1B by RNA interference and stable genomic deletion of exon 3b of *RAC1* resulted in upregulation of ALK5 expression [8], we addressed the question of whether this effect was direct or whether it involved the expression or activation of intermittent protein(s). In essence, we provide evidence for the latter possibility by identifying *F2RL1* as a gene whose expression (basal and TGF-β-dependent) is decreased by RAC1B. This reduction, in turn, results in a downregulation of ALK5 mRNA and protein expression (see Figure 2), which in turn reduces TGF-β1-dependent cell migration (see Figure 3) and growth inhibition (see Figure 4). This series of events was inferred from the observation that the increase in basal ALK5 expression in response to reducing the abundance of RAC1B was PAR2-dependent (see Figure 2A,B). However, the upregulation of PAR2 expression by knockdown of RAC1B, unlike the TGF-β1-dependent increase in PAR2 expression (see Figure 1B), was not affected by ALK5 siRNA transfection (see Figure 2D), indicating a TGF-β/ALK5-independent mechanism for PAR2 upregulation that is attenuated by RAC1B. Moreover, in Panc1 cells with RAC1B knocked down, the co-depletion of PAR2 prevented the upregulation of ALK5 (see Figure 2A), compromised the sensitivity of *TGFBR1* to upregulation by TGF-β1 (see Figure 2C), and rescued the cells from an increase in TGF-β1-dependent chemokinesis (see Figure 3A). Conversely, ectopic expression of HA-RAC1B was associated with reduced ALK5 expression (see Figure 1C and Appendix A) and a dramatically impaired sensitivity of *F2RL1* to upregulation of its mRNA by TGF-β1 stimulation (see Figure 1D). However, the impaired response of endogenous *F2RL1* to TGF-β1 stimulation in cells overexpressing RAC1B could be reversed by ectopic expression of PAR2, which partially protected these cells from the HA-RAC1B-induced decrease in basal and TGF-β1-dependent cell migration (see Figure 3B). Moreover, the knockdown of PAR2 alone (without concomitant knockdown of RAC1B) downregulated ALK5, leading to a reduction of TGF-β1-induced chemokinesis in HaCaT cells (see Appendix A) and growth inhibition in Panc1 (see Figure 4A) and HaCaT (see Appendix A) cells. Finally, we found that the RAC1B knockdown-induced rise in TGF-β1-induced PAR2 mRNA expression was sensitive to pharmacological inhibition of MEK-pERK but not MKK-p38 MAPK signaling (see Figure 5).

Surely, RAC1B-mediated inhibition of basal and TGF-β1-induced upregulation of PAR2 has still to be demonstrated at the protein level. If confirmed, it would be worth identifying the mechanism underlying negative regulation of basal (ALK5-independent) PAR2 expression by RAC1B (transcriptional or posttranscriptional, direct or indirect, Figure 6), and analyzing whether this RAC1 isoform decreases the sensitivity of cells to trypsin or PAR2 agonist-induced (TGF-β-independent) responses [10]. Since PAR2 has been identified here as a TGF-β target gene in pancreatic and skin cells and upregulation of its mRNA by TGF-β1/ALK5 was inhibited by RAC1B, ALK5 and PAR2 may form a positive feedback loop that is under negative control by RAC1B (Figure 6). Upon a decrease in RAC1B expression, the mutual stimulatory effects between ALK5 and PAR2 as well as induction of *TGFBR1* by TGF-β1 are freed from inhibition, eventually resulting in ALK5 and PAR2 upregulation. This, in turn, may amplify TGF-β/ALK5 or PAR2-dependent responses such as EMT, cell motility, and growth arrest, or proliferation, respectively.

It will be exciting to reveal whether this proposed network also operates in other tissues and cell types. In mouse mammary epithelial cells, Rac1b has been shown to be upregulated by MMP3 and to promote EMT. Interestingly, in a mouse mammary tumor model, TGF-β promoted the induction of EMT and translationally upregulated Mmp3 in an ALK5-dependent manner [15]. It is, hence, conceivable that TGF-β/ALK5-induced upregulation of Mmp3 (which likely requires PAR2) and subsequent induction of Rac1b by Mmp3 will eventually prevent an overshooting of TGF-β/ALK5 or PAR2-induced responses, thus adding negative feedback and an additional level of complexity to the system.

## 4. Material and Methods

### 4.1. Antibodies and Reagents

The following primary antibodies were used: Anti-HSP90, #sc-7947, #sc-13119, and anti-TGF-β receptor I (V22), #sc-398 (Santa Cruz Biotechnology, Heidelberg, Germany), anti-RAC1B, #09-271 (Merck Millipore, Darmstadt, Germany), anti-GAPDH (14C10), #2118, and anti-Myc-Tag (9B11), #2276 (Cell Signaling Technology, Frankfurt am Main, Germany), anti-HA (12CA5), #1583816 (Roche Diagnostics, Hague Road, IN, USA). HRP-linked anti-rabbit, #7074, and anti-mouse, #7076, secondary antibodies were from Cell Signaling Technology. Recombinant human TGF-β1, #300-023, was provided by ReliaTech (Wolfenbüttel, Germany) and used at a concentration of 5 ng/mL in all assays. Pharmacological inhibitors, SB203580, SB431542, and U0126, were obtained from Calbiochem/Merck (Darmstadt, Germany) and dissolved in dimethylsulfoxide to yield stock solutions of 10 mM.

### 4.2. Cells and Their Genetically Modified Derivatives

The human PDAC-derived cell lines Panc1, BxPC3, Colo357, and MiaPaCa2, and the immortalized human keratinocyte cell line, HaCaT, were originally obtained from the ATCC (Manassas, VA). All cell lines were maintained in RPMI 1640 supplemented with 10% fetal bovine serum, 1% Penicillin-Streptomycin-Glutamine (Life Technologies/Thermo Fisher Scientific, Waltham, MA, USA), and 1% sodium pyruvate (Merck Millipore). Prior to TGF-β treatment, cells were serum-starved (0.5% fetal bovine serum) for 12–16 h. The generation of individual, genetically homogenous clones of Panc1 cells with stable expression of HA-tagged RAC1B in the pCGN vector [2] and of Panc1 cells with a deletion of exon 3b of *RAC1* by CRISPR/Cas9 technology [8] has been described in detail previously. Counting of cells was done with a Neubauer chamber after detaching cells with trypsin/EDTA. Cell viability, as assessed by trypan blue exclusion, was >97%.

### 4.3. Transient Transfection of siRNA and Expression Vectors

For transient transfection of siRNA and DNA plasmids, cells were seeded on day 1 in Nunclon^TM^ Delta Surface plates (Nunc, Roskilde, Denmark) and transfected twice, on days 2 and 3, serum-free with 25 or 50 nM of siRNA specific for RAC1B, PAR2, or ALK5 [8,9], or the respective scrambled control siRNAs, for 4 h using Lipofectamine 2000 (Life Technologies). Additional validation of these siRNAs was performed for RAC1B [2], and ALK5 and PAR2 [8,9]. A PAR2 expression vector (PAR2-Myc-DKK) was purchased from Origene (Rockville, MD, USA).

### 4.4. QPCR Analysis

Total RNA was extracted from Panc1 cells using PeqGold RNAPure from Peqlab (Erlangen, Germany) and purified according to manufacturer’s instructions. For each sample, 2.5 μg RNA were subjected to reverse transcription for 1 h at 37 °C, using 200 U M-MLV Reverse Transcriptase and 2.5 μM random hexamers (Life Technologies) in a total volume of 20 μL. Relative mRNA expression of target genes was quantified by qPCR on an I-Cycler (BioRad, Munich, Germany) using Maxima SYBR Green Mastermix (Thermo Fisher Scientific, Waltham, MA, USA). Data were normalized to the expression of both TATA-box-binding protein (TBP) and β-actin. For sequences of PCR primers, see [8,9].

### 4.5. Immunoblotting

Cell lysis and immunoblotting was essentially performed as described previously [8]. In brief, cells were washed once with ice-cold PBS and lysed with 1× PhosphoSafe lysis buffer (Merck Millipore). Following clearance of the lysates by centrifugation, their total protein concentrations were determined with the DC Protein Assay (BioRad). Equal amounts of proteins were fractionated by polyacrylamide gel electrophoresis on mini-PROTEAN TGX any-kD precast gels (BioRad) and blotted to PVDF membranes. Membranes were blocked with nonfat dry milk or BSA and incubated with primary antibodies either for 2 h at RT or overnight at 4 °C. After washing and incubation with HRP-linked secondary antibodies, chemoluminescent detection of proteins occurred on a ChemiDoc XRS imaging system (BioRad) with Amersham ECL Prime Detection Reagent (GE Healthcare, Munich, Germany).

### 4.6. Migration Assays

We employed the xCELLigence^®^ DP system (ACEA Biosciences, San Diego, CA, USA, distributed by OLS, Bremen, Germany) as described previously [2] to measure chemokinesis of Panc1 and HaCaT cells. Briefly, CIM plates-16 were prepared according to the instruction manual, except that the underside of the upper chambers of the CIM plate-16 was coated with 30 μL of collagen I to facilitate adherence of the cells and enhance signal intensities. Following assembly of the chambers, filling of the lower chambers with medium (RPMI with 1% fetal bovine serum) chambers were equilibrated in the incubator for 1 h, The upper chamber of each well was loaded with 50,000 cells suspended in RPMI with 1% fetal bovine serum immediately after addition of TGF-β1 to the cell suspensions, while the lower chamber contained the same medium without cells. Data acquisition was done at intervals of 15 or 30 min and analyzed with RTCA software, version 1.2 (ACEA).

### 4.7. Statistical Analysis

Statistical significance was calculated using the unpaired two-tailed Student’s *t* test. Results were considered significant at *p* < 0.05 (*). Higher levels of significance were *p* < 0.01 (**) and *p* < 0.001 (***).

## 5. Conclusions

The data obtained in this study suggest that RAC1B is part of a regulatory network that negatively controls the cells’ sensitivity to TGF-β. Therapeutic targeting of this network at the level of RAC1B and/or PAR2 for inhibition or activation may represent a feasible strategy to enhance TGF-β’s tumor suppressor function or interfere with its malignant effects in fibrosis and tumor progression, respectively. 

## Figures and Tables

**Figure 1 cancers-11-01211-f001:**
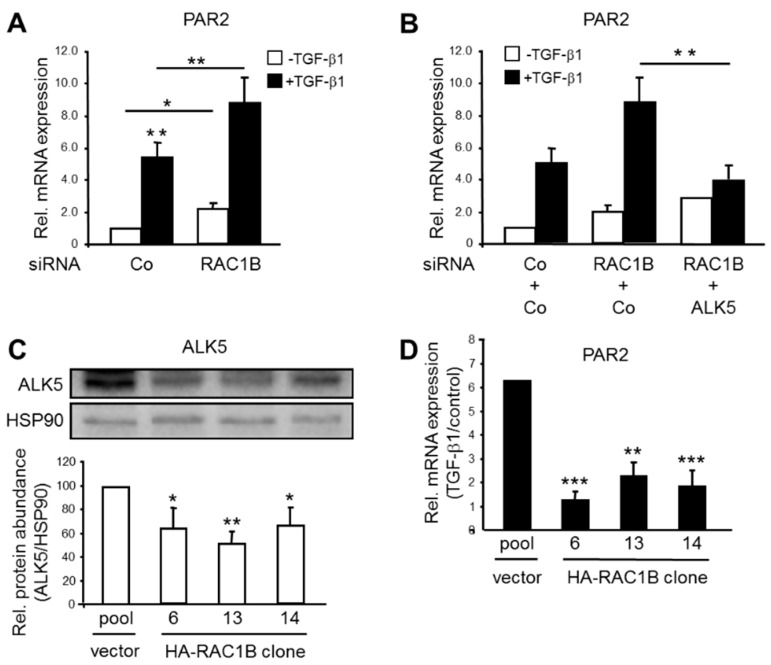
Effect of RAC1B knockdown or ectopic overexpression on TGF-β1-induced gene expression of PAR2. (**A**) Panc1 cells were transfected twice with 50 nM each of either control (Co) siRNA or RAC1B siRNA. Forty-eight hours after the second round of transfection, cells were treated with TGF-β1 for 24 h and subsequently subjected to qPCR analysis for PAR2. (**B**) As in (**A**), except that Panc1 cells were cotransfected with 25 nM each of Co siRNAs specific for RAC1B and ALK5, or 25 nM each of RAC1B siRNA + ALK5-specific Co siRNA, or 25 ng each of RAC1B siRNA + ALK5 siRNA prior to TGF-β1 treatment and qPCR analysis of PAR2. Data in (**A**) and (**B**) represent the normalized mean ± SD from three independent experiments and are displayed relative to control siRNA set arbitrarily at 1. (**C**) Three individual clones of Panc1 cells with stable ectopic expression of HA-tagged RAC1B (HA-RAC1B) were subjected to immunoblot analysis of ALK5. The graph below the immunoblot shows results from densitometry-based quantitative analysis. (**D**) As in (**C**) except that cells were treated, or not, with TGF-β1 for 48 h and subjected to qPCR analysis of PAR2. Data in (**C**) and (**D**) represent the normalized mean ± SD of three parallel wells from one representative experiment out of three experiments performed with very similar results. The asterisks indicate significance.

**Figure 2 cancers-11-01211-f002:**
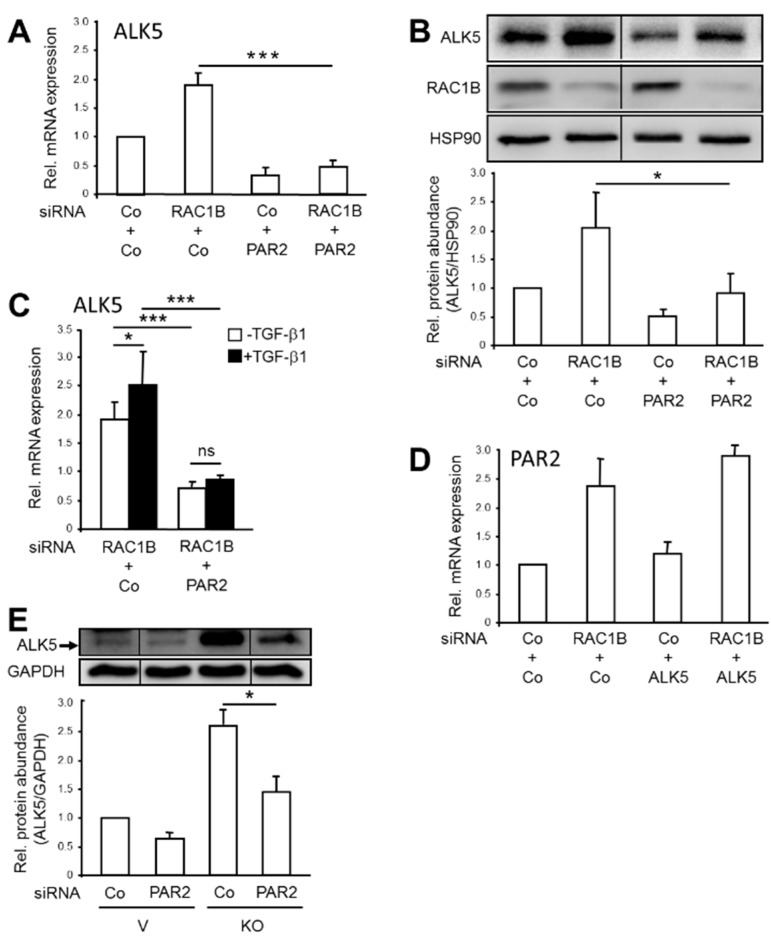
Effect of PAR2 knockdown on RAC1B knockdown-induced ALK5 expression in Panc1 cells. (**A**) Panc1 cells were transfected twice with 25 nM each of Co siRNAs specific for RAC1B and PAR2, 25 nM each of RAC1B siRNA + Co siRNA specific for PAR2, or 25 nM each of RAC1B siRNA + PAR2 siRNA. Forty-eight hours after the second transfection, these Panc1-RAC1B-KD cells were subjected to RNA isolation and qPCR analysis for ALK5. Data represent the normalized mean ± SD from three independent experiments and are displayed relative to control siRNA-transfected cells set arbitrarily at 1.0. (**B**) As in (**A**) except that RAC1B-knockdown cells were analyzed by immunoblotting for ALK5, RAC1B, and HSP90. The graph below the immunoblot shows results from densitometry-based quantitative analysis of ALK5. Data represent the mean ± SD from three experiments. (**C**) As in (**A**), except that the indicated transfectants were treated, or not, with TGF-β1 for 24 h prior to qPCR analysis of ALK5. (**D**) As in (**A**), except that cells were cotransfected with ALK5 siRNA instead of PAR2 siRNA and subjected to qPCR analysis of PAR2. (**E**) Panc1-RAC1B-KO cells (KO), and vector control cells (V) were transfected twice with 50 nM each of either Co siRNA or PAR2 siRNA. Forty-eight hours after the second transfection, the cells were subjected to immunoblot analysis of ALK5 and GAPDH. The asterisks in (**A**–**C**,**E**) indicate significance. The thin vertical lines in the immunoblot images of panels B and E indicate that irrelevant lanes have been removed.

**Figure 3 cancers-11-01211-f003:**
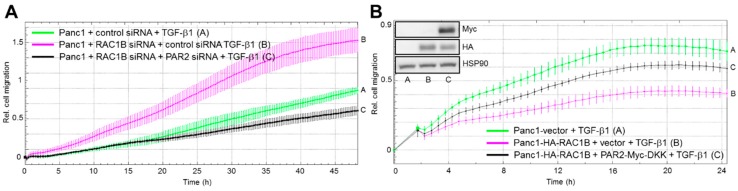
Effect of PAR2 depletion or ectopic overexpression on TGF-β-regulated chemokinesis in Panc1 cells with RAC1B knockdown or stable overexpression, respectively. (**A**) Panc1 cells were transfected twice with 25 nM each of Co siRNAs for RAC1B and PAR2, or 25 nM each of RAC1B siRNA and PAR2-specific Co siRNA, or 25 nM RAC1B siRNA + 25 nM PAR2 siRNA. Forty-eight hours after the second round of transfection, cells were assayed for migratory activity on an xCELLigence platform in the presence of TGF-β1. The graph shows a representative experiment. Data are the mean ± SD from 3–4 wells per condition. Differences between Panc1 + RAC1B siRNA + PAR2 siRNA + TGF-β1 (black curve, tracing C) and Panc1 + RAC1B siRNA + TGF-β1 (magenta curve, tracing B) are significant at 01:00 and all later time points. Successful inhibition of RAC1B and PAR2 was verified by immunoblotting and qPCR analysis, respectively. (**B**) Panc1 cells with ectopic expression of HA-RAC1B (clone 4) were transiently transfected with Myc-DKK-tagged PAR2, or empty vector, and 48 h later subjected to real-time cell migration assay in the presence of TGF-β1. Panc1 cells with stable expression of empty pCGN vector (Panc1-vector) rather than HA-RAC1B were used as control for the migration-inhibitory effect of HA-RAC1B. Shown is a representative experiment (mean ± SD from 3–4 wells per condition). Differences between Panc1-HA-RAC1B + PAR2-Myc-DKK + TGF-β1 (black curve, tracing C) and Panc1-HA-RAC1B + empty vector + TGF-β1 (magenta curve, tracing B) are significant at 04:00 and all later time points. Ectopic expression of HA-RAC1B and PAR2-Myc-DKK were verified in immunoblots using anti-HA and anti-Myc antibodies, respectively (inset).

**Figure 4 cancers-11-01211-f004:**
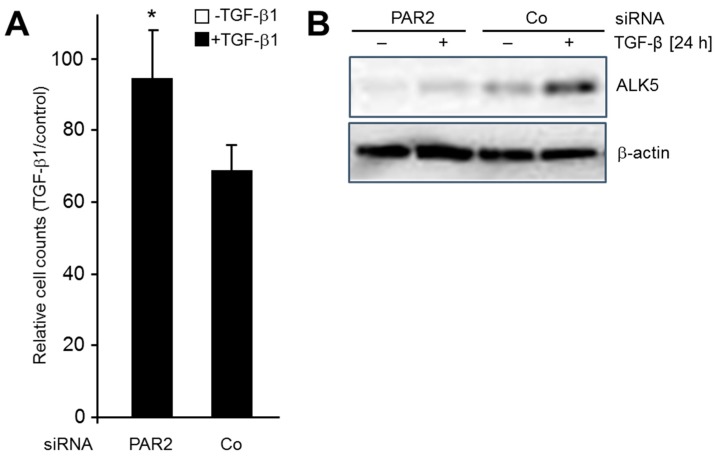
Effect of PAR2 knockdown on TGF-β1-induced growth inhibition in Panc1 cells. (**A**) Panc1 cells were transfected twice with 50 nM of either PAR2 siRNA or Co siRNA. Forty-eight hours after the second round of transfection cells were treated, or not, with TGF-β1 for 24 h, then detached and counted. Data are the mean ± SD of three experiments and are displayed as % reduction in cell numbers of TGF-β1-treated cells relative to numbers of untreated control cells. The asterisk indicates a significant difference. (**B**) As in (**A**) except that cells were lysed after TGF-β1 treatment and processed for immunoblotting of ALK5, and β-actin as a loading control.

**Figure 5 cancers-11-01211-f005:**
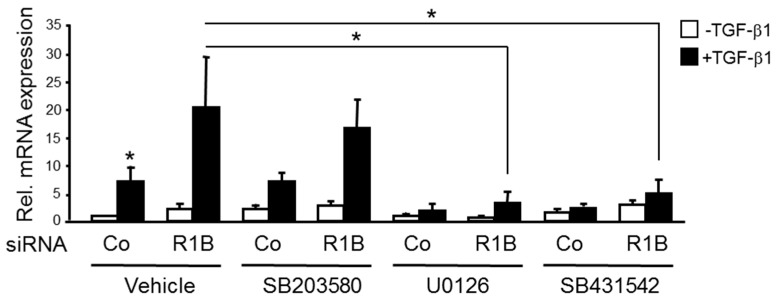
Effect of inhibition of MKK-p38 or MEK-ERK signaling on TGF-β1-induced regulation of PAR2 mRNA under conditions of a RAC1B knockdown. Panc1 cells were transfected twice with 50 nM each of either control (Co) siRNA or RAC1B (R1B) siRNA. Twenty-four hours after the second transfection, cells were serum-starved overnight and treated with vehicle (dimethylsulfoxide, 0.1%), SB203580 (10 μM), UO126 (10 μM), or SB431542 (5 μM). Thirty minutes after the addition of inhibitors, cells received TGF-β1 and were incubated for another 48 h in medium with 0.5% fetal bovine serum followed by qPCR analysis of PAR2. Data represent the normalized mean ± SD from three independent experiments and are displayed relative to Co siRNA-transfected, vehicle-treated cells set arbitrarily at 1. The asterisks indicate significant differences.

**Figure 6 cancers-11-01211-f006:**
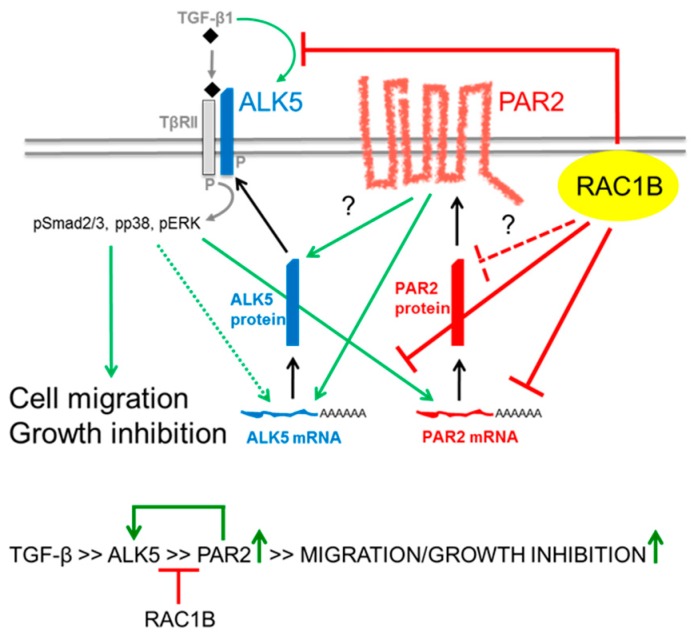
Schematic diagram illustrating RAC1B regulation of ALK5 via PAR2. TOP: RAC1B suppresses the abundance of basal and TGF-β1/ALK5-dependent mRNA (right, solid red lines) and protein (right, stippled red line) of PAR2. Reduction in the protein abundance of PAR2 leads to downregulation of ALK5 mRNA and, hence, protein abundance and/or membrane localization (center) and, thereby, of TGF-β1-induced Smad and non-Smad signaling (top left), culminating in reduction in TGF-β-induced PAR2 gene expression (bottom right) as well as cell migration and growth inhibition (bottom left). In addition, RAC1B suppresses autoinduction of *TGFBR1* by TGF-β1 (top). Stimulatory interactions are indicated by green arrows and inhibitory interactions by red lines. Stippled lines indicate the possibility that RAC1B targets PAR2 not directly but through intermediary proteins of as yet unknown identity, while the dotted arrow indicates that ALK5 autostimulation only occurs under conditions of RAC1B inhibition. The question marks indicate possible effects on protein stability independent of transcriptional events. P, phosphate residue. BOTTOM: Summary of effect: RAC1B attenuates (red lines) upregulation of PAR2 by activated ALK5 (upward green arrow), which in turn affects the upregulation of ALK5 (green line/arrow). The RAC1B inhibitory effect on basal PAR2 expression, which is ALK5-independent (see Figure 2D), has been omitted here for reasons of clarity.

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
