# Peer review of "RAC1B Suppresses TGF-β-Dependent Chemokinesis and Growth Inhibition through an Autoregulatory Feed-Forward Loop Involving PAR2 and ALK5"

_cancers, 2019, doi:10.3390/cancers11081211_

Round 1

Reviewer 1 Report

Major findings

1. The authors need to perform rescue experiments where the RAC1B is re-introduced after knockdown of RAC1B and probe for PAR2 and ALK5 expression at protein level.

2. The authors should consider using Mouse embryo fibroblasts of RAC1B to check for the expression of PAR2 and ALK5.

3. The Figure 1 mRNA data should be supported with Immunoblots.

4. The authors should consider doing an experiment to check if ALK5 is regulating PAR2 expression or vice versa by transfecting increasing amounts of ALK5 or PAR2 and probe for the respective proteins. 

5. Whether this process is involved in EMT of cancer cells. The authors could check for Epithelial or mesenchymal markers by qPCR.

6. The authors should check for F2RL1 expression and try to correlate their findings.

7. Pictures of Migration or Invasion assays would be more informative

Minor findings

Please indicate the name of the gene in the figure panel as it easy to follow.

Author Response

Dear Editor:

We thank the reviewers for their generally very positive comments on our manuscript and have done our best to incorporate their suggestions in the revised version. Changes and additions to the original version have been highlighted in the “track changes” mode. Specifically, we would like to respond to the reviewer’s comments as follows:

Reviewer 1

Major findings

The authors need to perform rescue experiments where the RAC1B is re-introduced after knockdown of RAC1B and probe for PAR2 and ALK5 expression at protein level.

Response: We thank the reviewer for suggesting this good control experiment. Since we did not have available a mutant form of RAC1B resistant to siRNA-mediated degradation, we have re-introduced by transient transfection HA-tagged RAC1B into Panc1-CR-RAC1B-KO cells. We found that ectopic RAC1B rescued Panc1 cells from the increase in ALK5 abundance (detected by immunoblotting) and PAR2 mRNA expression (detected by qPCR). These data are shown in Supplementary Figure S1C. As outlined in the manuscript, PAR2 could not be analyzed at the protein level due to lack of suitable antibodies. 

The authors should consider using Mouse embryo fibroblasts of RAC1B to check for the expression of PAR2 and ALK5.

Response: This is certainly a good idea, however, we would like to stay in a human cancer-related cellular system. Apart from this, obtaining all the necessary (mouse-specific) reagents and probes would be very time-consuming.

The Figure 1 mRNA data should be supported with Immunoblots.

Response: As requested, we have supplied an immunoblot for ALK5 in panel C. This blot has replaced the ALK5 mRNA data which are now shown in the Supplementary data file as Figure S1B. As mentioned above, PAR2 cannot be detected by immunoblotting.

The authors should consider doing an experiment to check if ALK5 is regulating PAR2 expression or vice versa by transfecting increasing amounts of ALK5 or PAR2 and probe for the respective proteins.

Response: ALK5 does not regulate PAR2 since siRNA-mediated knockdown of ALK5 (or ectopic expression of a kinase-dead ALK5 mutant (ALK5-K232R) did not reduce PAR2 mRNA expression but reversed the RAC1B knockdown-induced increase in TGF-β1-induced upregulation of PAR2 mRNA (see Figure 1B). This was confirmed by transfecting increasing amounts of ALK5 which failed to alter basal PAR2 mRNA expression, while, in contrast, transfecting increasing amounts of PAR2 did increase ALK5 protein abundance. However, since the dependency of ALK5 expression on PAR2 expression has been analyzed by us in detail in a previous publication (Zeeh et al. 2016), we prefer not to show these data here as this would not add novel information to the manuscript. 

Whether this process is involved in EMT of cancer cells. The authors could check for Epithelial or mesenchymal markers by qPCR.

Response: We have extensively characterized negative regulation of TGF-b1-induced EMT by RAC1B in a previous publication (see Ref. 3 in the revised version). RAC1B promotes E-cadherin expression and inhibits expression of Slug, Snail, and several other mesenchymal marker proteins. It would, indeed, be exciting to analyze if RAC1B regulation of these EMT markers is PAR2-dependent. Our demonstration that PAR2 is involved in RAC1B-mediated suppression of TGF-b1/ALK5-mediated cell migration (which is a EMT-associated process) argues in favor of such a role for PAR2. However, we believe that analyzing the role of PAR2 in RAC1B-mediated suppression of TGF-b1-induced EMT is beyond the scope of this work.

The authors should check for F2RL1 expression and try to correlate their findings.

Response: We do not really know where to check for PAR2 expression (the sentence does not say). If it relates to EMT it would be nice to study the role of PAR2 in EMT, however, this would require a large study and is beyond the scope of the present study.

Pictures of Migration or Invasion assays would be more informative

Response: We have chosen this assay type because the migration data produced on the xCELLigence platform are highly accurate and quantitative. Additional advantages of this technique are that it does not require labeling of the cells and that it does monitor the cells’ migratory activities in real-time. Alternatively, the assays could be done with scratch assays in combination time-lapse microscopy, but, unfortunately we do not have available the necessary equipment (including the analysis software for data quantification) in our department.

Please indicate the name of the gene in the figure panel as it easy to follow.

Response: Done

Reviewer 2 Report

In this manuscript, the authors elaborate on previously published data on the role of RAC1b in pancreatic cancer cells, which have established RAC1b as an inhibitor of TGF-β1-induced epithelial-mesenchymal transition. Previous work had already shown that RAC1B downregulates the TGF-β receptor ALK5 and that the G protein-coupled receptor PAR2 protein was required for maintaining expression of ALK5 and for the pro-oncogenic effects of TGF-β1.Here the authors describe that RAC1b targets the expression of PAR2, which then affects ALK5 levels. They further show that PAR2 itself is a TGF-β-upregulated gene under positive control by ALK5, suggesting that RAC1b interferes with a positive regulatory loop modulating TGF β-induced cell motility and growth inhibition.

Concerns and suggestions:

1) Compared to the data already published by the same authors, this manuscript identifies PAR2 as a target for the effects of RAC1b but does not add substantially novel information. The data are of more descriptive nature and do not clarify the underlying mechanism. I thus feel this study is incomplete and requires some further details to get more impact.

Possible additional experiments, as in part already discussed by the authors, could include studies to clarify (a) whether RAC1b affects PAR2 gene transcription using a PAR2 promoter-driven construct; (b) whether RAC1b affects PAR2 expression by interfering with any of the previously analyzed pathways (e.g. pERK, p38 MAPK, ROS production);

2. Considering the important previous studies on the role of RAC1B in pancreatic and breast cancer cells, it is unclear why the authors in this study included HaCaT keratinocytes and not breast cancer cells.

a) The impact of the results would be much higher if the study would show the observed effects also in breast cancer cells. This is especially relevant since other reports have presented contradicting data and concluded that RAC1b has a pro-EMT effect in breast cancer.

b) Besides this experimental suggestion, the presented conclusion that RAC1B is part of a regulatory network that negatively controls the cells’ sensitivity to TGF-b, should be discussed concerning the state of this network in breast cancer cells, and if or how the published MMP3-treatment of epithelial breast cells may affect this network.

other comments

3) The state of the art description in the introduction is documented by only a few citations from the authors’ own publications and does not appropriately acknowledge the original literature reports from various researchers in the field. Their contribution should be cited because bibliographic metrics, including the number of citations are, after all, an internationally used measure of an author’s or journal’s research impact.

4) line 50: the sentence “Previous data from our group also indicated that RAC1B suppresses expression, kinase activity, and TGF-b1-mediated upregulation of ALK5, (…)” is misleading as it may suggest a direct effect of RAC1b on kinase activity. This is not the case and the observation described in the cited paper was that an increase in kinase activity was merely associated with an increase in ALK5 protein abundance.

5) line 206: “Together, these data show that downregulation of PAR2 by RAC1B results in a decrease in ALK5 abundance which (…); the part ‘by RAC1b’ is not shown in these experiments and cannot be concluded from this paragraph, only that downregulation of PAR2 alone is sufficient for the observed decrease.

6) the authors employ rather complicated sentence structures due to double-negative expressions ( i.e. inhibition of inhibitory effects). An example is the subheading:

2.3. RAC1B Knockdown-Mediated Upregulation or RAC1B Overexpression-Mediated Downregulation of TGF-b1-induced Cell Migration is Reversed by PAR2 Knockdown or Overexpression, Respectively

I suggest simplifying some sentences or splitting them into two in order to facilitate comprehension upon Reading.

7) minor correction in the references, citations 4 and 5 by Melzer et al. are labeled as 2017a and 2017b, respectively.

Author Response

Dear Editor:

We thank the reviewers for their generally very positive comments on our manuscript and have done our best to incorporate their suggestions in the revised version. Changes and additions to the original version have been highlighted in the “track changes” mode. Specifically, we would like to respond to the reviewer’s comments as follows:

Reviewer 2

Comments and Suggestions for Authors

In this manuscript, the authors elaborate on previously published data on the role of RAC1b in pancreatic cancer cells, which have established RAC1b as an inhibitor of TGF-β1-induced epithelial-mesenchymal transition. Previous work had already shown that RAC1B downregulates the TGF-β receptor ALK5 and that the G protein-coupled receptor PAR2 protein was required for maintaining expression of ALK5 and for the pro-oncogenic effects of TGF-β1.Here the authors describe that RAC1b targets the expression of PAR2, which then affects ALK5 levels. They further show that PAR2 itself is a TGF-β-upregulated gene under positive control by ALK5, suggesting that RAC1b interferes with a positive regulatory loop modulating TGF β-induced cell motility and growth inhibition.

Concerns and suggestions:

1) Compared to the data already published by the same authors, this manuscript identifies PAR2 as a target for the effects of RAC1b but does not add substantially novel information. The data are of more descriptive nature and do not clarify the underlying mechanism. I thus feel this study is incomplete and requires some further details to get more impact.

Possible additional experiments, as in part already discussed by the authors, could include studies to clarify (a) whether RAC1b affects PAR2 gene transcription using a PAR2 promoter-driven construct; (b) whether RAC1b affects PAR2 expression by interfering with any of the previously analyzed pathways (e.g. pERK, p38 MAPK, ROS production).

Response: We agree with the reviewer that identifying the pathway(s) that are involved in negative regulation of PAR2 by RAC1b is an issue worth analyzing. We used inhibitors of MEK-ERK (U0126), MKK-p38 (SB203580) and ALK5 (SB431542) signaling and found that only U0126 (and SB431542 as control) was able to block the RAC1B knockdown-mediated increase in TGF-b1-induced PAR2 mRNA expression. These data are shown in the new Figure 5 in the revised version.

Considering the important previous studies on the role of RAC1B in pancreatic and breast cancer cells, it is unclear why the authors in this study included HaCaT keratinocytes and not breast cancer cells.

Response: An explanation for why HaCaT cells have been used is given in section 2.1, first paragraph. We found earlier that in PDAC-derived cells and in HaCaT cells PAR2 is required for sustaining ALK5 expression (see Zeeh et al. 2016), but whether this also applies to breast cancer cells is not known. An additional reason for using HaCaT cells is their high sensitivity to TGF-b as evidenced by their strong migratory activity and growth arrest induced by TGF-β1 (see Supplementary Figures S3 and S4, respectively).

a) The impact of the results would be much higher if the study would show the observed effects also in breast cancer cells. This is especially relevant since other reports have presented contradicting data and concluded that RAC1b has a pro-EMT effect in breast cancer.

Response: Rac1b has pro-EMT effect for MMP3 as EMT inducer but TGF-β-induced EMT in breast cancer cells has not yet been analyzed. The reason why we did not include breast cancer cells are: i). The PAR2 dependency of ALK5 expression has not yet been studied in breast cancer cells (see above) ii) Most breast cancer cell lines including MDA-MB-231 as well as normal human mammary epithelial cells (HMEC, see migration data in Melzer et al. 2017a) are not or only weakly responsive to TGF-β with respect to migration, and ii) at least MDA-MB-231 cells - in our hands - do not, or only marginally, upregulate PAR2 mRNA in response to a 24-h TGF-β1 treatment. We therefore prefer to focus on PDAC-derived cells in this study and if using non-PDAC cells, then only those in which ALK5 regulation by PAR2 has been documented, i.e. HaCaT. In order to demonstrate that TGF-β regulation of PAR2 is a general feature in pancreatic epithelial cells, we have tested additional PDAC-derived cell lines, Colo357 and BxPC3, and found that both responded to TGF-β1 stimulation with upregulation of PAR2 mRNA, albeit not to the extent seen in Panc1 cells. These data have been included in Supplementary Figure S1A.

b) Besides this experimental suggestion, the presented conclusion that RAC1B is part of a regulatory network that negatively controls the cells’ sensitivity to TGF-b, should be discussed concerning the state of this network in breast cancer cells, and if or how the published MMP3-treatment of epithelial breast cells may affect this network.

Response: Since in breast cancer cells, data on the role of Rac1b regulation of ALK5 and PAR2 are scarce (which is in part due to the poor TGF-β sensitivity of the available human cell lines, see above) it is difficult to compare this network with that in PDAC-derived cells. However, we have speculated on how this network may be impacted by MMP3 based on what is known about the interactions between MMP3 and TGF-β (see last paragraph of the Discussion section).

other comments

3) The state of the art description in the introduction is documented by only a few citations from the authors’ own publications and does not appropriately acknowledge the original literature reports from various researchers in the field. Their contribution should be cited because bibliographic metrics, including the number of citations are, after all, an internationally used measure of an author’s or journal’s research impact.

Response: We, of course, agree with this issue. It should be stated, however, that the role of Rac1b in TGF-β signaling has only been studied by our group. Nevertheless, we have broadened the discussion and have cited relevant literature to acknowledge the contributions from other colleagues. In this context, we should like to point out that we have recently published a review article that does include nearly all original literature reports on Rac1b (see Ref. 1, Melzer et al. 2019). At several occasions, we have cited this review rather than the original work from others in order to keep the number of cited references at a minimum.

4) line 50: the sentence “Previous data from our group also indicated that RAC1B suppresses expression, kinase activity, and TGF-b1-mediated upregulation of ALK5, (…)” is misleading as it may suggest a direct effect of RAC1b on kinase activity. This is not the case and the observation described in the cited paper was that an increase in kinase activity was merely associated with an increase in ALK5 protein abundance.

Response: This is correct and we have rectified this in the revised version.

5) line 206: “Together, these data show that downregulation of PAR2 by RAC1B results in a decrease in ALK5 abundance which (…); the part ‘by RAC1b’ is not shown in these experiments and cannot be concluded from this paragraph, only that downregulation of PAR2 alone is sufficient for the observed decrease.

Response: This is correct and we have modified the sentence accordingly.

6) the authors employ rather complicated sentence structures due to double-negative expressions ( i.e. inhibition of inhibitory effects). An example is the subheading:

2.3. RAC1B Knockdown-Mediated Upregulation or RAC1B Overexpression-Mediated Downregulation of TGF-b1-induced Cell Migration is Reversed by PAR2 Knockdown or Overexpression, Respectively

I suggest simplifying some sentences or splitting them into two in order to facilitate comprehension upon Reading.

Response: We agree with this point of critizism. In order to facilitate understanding, we have rephrased the above mentioned sentence. However, we do not want to remove the double-negative expressions throughout the manuscript because it was our intention to express in the subheadings of the Results section what we actually did (i.e. knocking down or overexpressing RAC1B) and what the outcome of the respective manipulation was. This contrasts with the conclusions which we always expressed in a positive form. 

7) minor correction in the references, citations 4 and 5 by Melzer et al. are labeled as 2017a and 2017b, respectively.

Response: The citations refer to two different publications from the same year. In order to identifiy them as such we have labeled them as “a” and “b”.

Reviewer 3 Report

In this study, the authors investigated how RAC1B down-regulates ALK5. Since PAR2 is required for sustaining ALK5 expression, the authors examined if PAR2 is involved in RAC1B-dependent suppression of ALK5. The authors found that PAR2 and ALK5 form a feed-forward loop, where RAC1B negatively regulates transcription of PAR2, leading to suppression of this feed-forward loop.

This study has been performed reasonably and this review has no major comments.

Other comments:

1.    In Figure 1A and 1B, it is better to indicate what mRNA levels were examined like Figure 1C and 1D.

2.    In Figure 1D, while Rel. mRNA expression (TGF-beta1/control) is plotted along the y-axis, open and closed bars for “-TGF-beta1” and “+TGF-beta1” are indicated.

3.    Typo at line 100: “Figure 2”-> F2RL1or PAR2 ?

4.    Type at line 166: “… TGF- 1/ALK5-mediated upregulation of PAR2 and(see Figure 1C, D)”

Author Response

Dear Editor:

We thank the reviewers for their generally very positive comments on our manuscript and have done our best to incorporate their suggestions in the revised version. Changes and additions to the original version have been highlighted in the “track changes” mode. Specifically, we would like to respond to the reviewer’s comments as follows:

Reviewer 3

In this study, the authors investigated how RAC1B down-regulates ALK5. Since PAR2 is required for sustaining ALK5 expression, the authors examined if PAR2 is involved in RAC1B-dependent suppression of ALK5. The authors found that PAR2 and ALK5 form a feed-forward loop, where RAC1B negatively regulates transcription of PAR2, leading to suppression of this feed-forward loop.

This study has been performed reasonably and this review has no major comments.

Other comments:

In Figure 1A and 1B, it is better to indicate what mRNA levels were examined like Figure 1C and 1D.

Response: Done

In Figure 1D, while Rel. mRNA expression (TGF-beta1/control) is plotted along the y-axis, open and closed bars for “-TGF-beta1” and “+TGF-beta1” are indicated.

Response: We thank the reviewer for drawing our attention to this error. We have removed the legend in the figure.

Typo at line 100: “Figure 2”-> F2RL1or PAR2 ?

Response: “PAR2”. This typo was apparently introduced during editorial typesetting and has been removed.

Type at line 166: “… TGF- 1/ALK5-mediated upregulation of PAR2 and(see Figure 1C, D)”

Response: Rectified.

Round 2

Reviewer 2 Report

The authors have satisfactorily addressed the main concerns